# Amphimeriasis in Ecuador—Prevalence, Knowledge, and Socio-Cultural Practices Among Indigenous Chachi and Montubios Populations: A Mixed-Methods Cross-Sectional Study

**DOI:** 10.3390/tropicalmed9100248

**Published:** 2024-10-20

**Authors:** Richar Rodríguez-Hidalgo, William Cevallos, Maritza Celi-Erazo, Verónica Vargas-Roman, Luvin Oviedo-Racines, José Buitrón, Ayelén Lema, Manuel Calvopina

**Affiliations:** 1Instituto de Investigación en Zoonosis, Universidad Central del Ecuador, Quito 170521, Ecuador; rrodriguez@uce.edu.ec (R.R.-H.); mceli@uce.edu.ec (M.C.-E.); 2Facultad de Medicina Veterinaria y Zootecnia, Universidad Central del Ecuador, Quito 170521, Ecuador; jlbuitron@uce.edu.ec (J.B.); aalema@uce.edu.ec (A.L.); 3Programa de Doctorado en Salud Pública, Universidad Iberoamericana, Cancún 01219, Mexico; 4Facultad de Medicina, Universidad Central del Ecuador, Quito 170521, Ecuador; wcevallos@uce.edu.ec; 5Department of Social and Cultural Anthropology, KU Leuven, 3000 Leuven, Belgium; verova@gmail.com,; 6Department of Dermatology, Amsterdam University Medical Centers (UMC), 1105 AZ Amsterdam, The Netherlands; 7Infectious Diseases Programme, Amsterdam Institute for Infection and Immunology, Amsterdam University Medical Centers (UMC), 1105 AZ Amsterdam, The Netherlands; 8Área de Soberanía Alimentaria, Dirección Gestión de Fomento y Desarrollo Productivo, Gobierno Autónomo Descentralizado de la Provincia Esmeraldas, Esmeraldas 080102, Ecuador; loviedoracines@gmail.com; 9One Health Research Group, Facultad de Medicina, Universidad De Las Américas (UDLA), Quito 170124, Ecuador

**Keywords:** *Amphimerus*, amphimeriasis, KAP, ethnic groups, Ecuador

## Abstract

*Amphimerus*, a liver fluke, is the causative agent of amphimeriasis, a foodborne disease acquired thought the consumption of infected raw or undercooked river fish—a practice embedded in traditional culinary customs. Amphimeriasis represents a significant public health issue and has been endemic in Ecuador since 2011, particularly among the Chachi Amerindians and Montubios populations residing in tropical ecoregions. By employing a mixed-methods research design, we conducted a community-based, cross-sectional study. A survey comprising of 63 questions on KAP was administrated in person to community members, health personnel, and academic staff in the two populations. Additionally, 67 semi-structured interviews were performed. Microscopy was achieved on 273 human and 80 dog fecal samples to detect *Amphimerus* eggs. A total of 86 questionnaires (54 Chachi) and 67 interviews (44 Chachi), out of 300 residents, were completed. Among the respondents, 31.4% were aware of *Amphimerus*, locally referred to as “liver worm”. Although 79.1% reported not consuming raw fish, most admitted eating raw fish with lime juice and salt, a preparation known as “curtido”, and 59.3% reported consuming smoked fish. Here, 86.1% of participants considered “liver worm” a serious disease, and 55.8% recognized raw or marinated fish as a potential transmission route. The Chachi showed a preference for smoked fish, whereas the Montubios favoured “curtido”. The prevalence of *Amphimerus* infection was 23% in humans and 16.2% in dogs. Differences in KAP were observed between infected and non-infected individuals. Local health and academic personnel demonstrated insufficient knowledge about amphimeriasis. Some religious individuals refrained from participating, stating that they were “with God”. Despite the high prevalence of *Amphimerus* infection in both humans and dogs, knowledge about the parasite, the disease, and its transmission routes remains limited. Health education initiatives should be designed to modify the population’s KAP. It is crucial for national and local health authorities, as well as religious leaders, to be informed and actively involved in the prevention and control of amphimeriasis.

## 1. Introduction

Infectious agents, including viruses, bacteria, and parasites, often become entangled with the cultural practices and habits of the affected populations, shaping transmission patterns and healthcare-seeking behaviors. The knowledge, attitudes, and practices (KAP) of individuals in communities endemic to communicable diseases, alongside various socio-demographic and cultural factors, are widely recognized as critical determinants influencing the persistence, spread, and effectiveness of control or elimination efforts for these pathogens.

Food habits, including dietary choices and food preparation practices, can significantly influence the risk of contracting infectious diseases by acting as a vector for pathogens. In the case of foodborne trematodiasis, which is classified as a Neglected Tropical Disease by the WHO, food habits—particularly the consumption of raw or undercooked freshwater fish, crustaceans, and aquatic plants—are closely associated with the transmission of these parasites [1].

Foodborne trematodes (flukes) include the parasites *Clonorchis* [2], *Opisthorchis* [3], *Fasciola* [4] and *Paragonimus* [5]. These parasitic flukes have a complex life cycle that involves one or two intermediate hosts, and multiple definitive hosts and reservoirs, classifying them as zoonotic pathogens. Infections caused by these parasites can result in severe liver and lung diseases, collectively contributing to an estimated two million life years lost to disability and death worldwide each year [1]. The liver trematodes of the Opisthorchiidae family include *Clonorchis sinensis*, *Opisthorchis* spp., and *Amphimerus* spp. These liver flukes are acquired by consuming raw or undercooked freshwater fish, a practice associated with traditional fish recipes, leading to “fish-borne trematodiasis” [6,7]. The transmission of fish-borne trematodiasis involves various stages of the parasite infecting different intermediate hosts, such as snails and fish, before being ingested by humans. This transmission is closely linked to human behavioral patterns, particularly methods of producing, processing, and preparing foods. The traditional customs of consuming raw or undercooked freshwater fish, where the infective parasitic form, metacercaria, is present, play a significant role in the spread of these parasites [8]. Different population groups employ unique methods for preparing fish, and cultural beliefs also influence the transmission and endemicity of fish-borne trematodiasis, even within the same country [8]. For instance, “Ceviche”—a traditional Latin American cuisine made from fresh raw fish, oysters, or shrimp marinated in lime juice and salt—is a traditional dish popular from Mexico to Chile.

*Amphimerus* is a liver trematode of global distribution, infecting various mammals and freshwater fish-eating birds [9,10]. The importance of amphimeriasis as a public health problem has become increasingly recognized since it was reported as endemic in Ecuador [11,12]. In Ecuador, human infections were first diagnosed in 2011 among the Chachi indigenous group residing in the rural tropical rainforest of Esmeraldas province, in the northwest of the country. Subsequently, a new focus of infection was discovered in the rural tropical dry forest of Manabí province, located south of Esmeraldas, within a Montubios population [11]. Amphimeriasis is considered a zoonotic disease acquired through the consumption of raw or undercooked river fish [11,12,13]. The life cycle of *Amphimerus* involves reservoirs (humans, dogs, and cats) and intermediate hosts (snails and fish) that develop in the freshwater bodies of tropical regions. In Ecuador, four edible freshwater fish species have been identified as sources of *Amphimerus* infection [13], and the snail *Aroapyrgus* was found to be infected, serving as the first intermediate host [14]. Additionally, dogs and cats exhibited a high prevalence of *Amphimerus* infection in both endemic areas [12]. Preliminary observations indicate that the Chachi group habitually consumes smoked fish [11,15,16], while the Montubios prefer raw fish prepared as “marinated” or “curtido”, dishes in which fish is immersed in lime juice [13]. In both regions, a high prevalence of human infection has been reported, ranging from 26% to 36% [11,17].

We hypothesized that the behavioral–psychosocial background, occupation, and level of education of the infected individuals might play a role in the endemicity and persistence of *Amphimerus* infection. Given the importance of amphimeriasis as a public health problem, cultural and social factors specific to different ethnic groups may influence transmission dynamics and infection. Therefore, we designed and conducted the present study to assess the levels of knowledge, attitudes, and practices (KAP), as well as the socio-cultural beliefs of two endemic populations, using a mixed cross-sectional method. The aim is to understand the behaviors of both groups in relation to *Amphimerus* epidemiology, to implement effective intervention strategies for the prevention, control, and/or elimination of *Amphimerus* infection. Additionally, stool samples were microscopically analyzed to update the information on infection prevalence.

## 2. Materials and Methods

### 2.1. Study Sites

The study was conducted at two sites in the Pacific coastal region of Ecuador, between February and November 2023. These locations were selected based on previous studies of *Amphimerus* infection endemicity: (1) communities of the Chachi Amerindians in the northwestern region of Ecuador, in Esmeraldas province (coordinates: 0.746353, −78.919150; approximately 110 km from the Pacific Ocean) and (2) Montubios communities in the central Pacific Coastal region of Ecuador, in Manabí province (coordinates: −1.704417, −80.561539; approximately 60 km from the Pacific Ocean). The climate in the Chachi communities is characterized by tropical rainforest conditions with large rivers, while the Montubios communities experience a tropical dry climate with seasonal water streams. Both areas are marked by poverty, vulnerability, gender inequalities, and limited or non-existent sanitary conditions. The Chachi communities are accessible only by canoe. Additionally, dogs and cats in these areas are free roaming (Figure 1) [18].

### 2.2. Study Populations

According to the most recent census, the combined population of both groups is approximately 300 inhabitants in the studied communities. The Chachi indigenous people (Figure 2) are an ethnic group with a family-based social organization and communal decision-making, where local leaders play a crucial role in resolving conflicts and guiding decision-making processes. The Chachi primarily speak “chá palaa”, an independent language, although many also speak Spanish. Fishing in nearby rivers and streams is a significant activity for their daily sustenance, particularly for children and adolescents. The Chachi engage in subsistence agriculture, hunting, fishing, and gathering fruits and wild plants. They cultivate crops such as bananas, cassava, corn, and cocoa. The Chachi have a cultural tradition of smoking meat and fish obtained through hunting and fishing. Their rich cultural heritage includes music, dance, mythology, and religious ceremonies. They celebrate important festivals such as Chapi Malu (Christmas), Avemalu or Dyusa Malu (Easter), and Matsunu Fandangu (marriage) with traditional music, dance, and rituals. The Chachi people hold animistic and shamanistic beliefs, worshiping nature and ancestral spirits. However, due to the influence of missionaries in the region, many have also embraced Christianity, including Catholicism [13,14].

The Montubios are part of the ethnic group known as the “Mestizos”. Their identity is deeply rooted in a rural and agricultural environment, enriched by cultural traditions that encompass religious beliefs and ancestral practices. This identity is reflected in their customs, agricultural practices, and diet, which blend indigenous, Afro-Ecuadorian, and European elements [19]. All members of the Montubios community speak Spanish. Food and cooking methods are central to the “cholo manaba” culture. Common dishes include “ceviches”—prepared with raw fish, shrimp, or shellfish marinated in lime juice. The Montubios engage in medium-scale agriculture and livestock farming, particularly cultivating crops such as corn and coffee. Fishing, primarily during the drier months from July to October, serves both as a means of subsistence and a recreational activity. In the studied communities, a traditional preparation involves marinating raw sea and river fish with lime juice and salt, a dish known as “curtido” (Figure 3).

### 2.3. Study Design

A mixed-type observational and analytical epidemiological study was conducted, combining a knowledge, attitudes, and practices (KAP) questionnaire (quantitative) with semi-structured in-depth interviews (qualitative). The quantitative approach assessed the levels of KAP among participants, while the qualitative approach explored the underlying reasons and motivations behind these KAPs. The combination of both methods provided comprehensive and up-to-date coverage of the studied population, enabling a detailed and in-depth exploration of the issue. Participants included residents (male and female, over 18 years old), as well as health and academic sector personnel who agreed to participate. The KAP questionnaire consisted of 63 questions covering geographical and demographic variables, knowledge, attitudes, and practices related to *Amphimerus* and amphimeriasis, treatment, fish preparation and consumption, as well as water and sanitation.

All community members were invited to participate in the study by providing a fecal sample and collaborating in the quantitative questionnaire and qualitative interview. Each household head was provided with a plastic stool collector containing 10% formalin and given instructions on how to collect the fecal samples. In the following days, we collected the samples along with information on the age and gender of the participants. Fresh dog fecal samples (approximately 5 g) were collected from the ground early in the morning by the researchers. The stool samples were preserved in 10% formalin and transported at ambient temperature to research laboratories in Quito. Each sample was microscopically analyzed for the presence of *Amphimerus* eggs using the formalin—ether concentration method [20]. Quantitative data were collected using the “Epicollect5” application—a centralized platform that facilitated the cleaning and anonymization of participant data.

### 2.4. Statistical Analysis

The variables “awareness”, corresponding to the question “Do you know, or have you heard of liver fluke worm?”, and “ethnicity”, related to the question “Which ethnicity do you belong to?”, were considered the dependent variables for the analysis of amphimeriasis. A specific analysis was conducted on the 27 respondents who answered affirmatively to being aware of amphimeriasis. Additionally, a subset of 31 questionnaire participants categorized as “infected/non-infected”, based on coprological test results, was created. The relationship between these dependent and the independent variables was examined using descriptive statistics, Chi-square association tests, and Fisher’s exact tests, utilizing the “R” Project 4.4.2 software. The qualitative data obtained from the in-depth interviews were recorded and transcribed in Spanish to capture opinions and perspectives related to *Amphimerus* and amphimeriasis. The interviews were conducted with male and female participants over the age of 18 who voluntarily shared their insights, allowing the researcher to explore the topic in greater depth. For the Chachi participants, local translators were employed for the “chá palaa” language. The semi-structured interviews were analyzed using NVivo 14 (version 14.23.2) software, which facilitated the assignment of free codes to the open-ended responses and helped identify categories relevant to the study. These codes were continuously compared until patterns and relationships emerged, leading to the identification of significant categories.

### 2.5. Ethical Considerations

The study protocol was approved by the Ethics Committees of the Universidad Central del Ecuador (Code: 005-S-FMVZ-2022) and the Universidad Internacional Iberoamericana (Code: CR-172). During the initial meeting, community leaders and household heads were informed about the study and invited to participate. Signed approval was obtained from community leaders and local health authorities before data collection and sampling. Written informed consent was obtained from all participants prior to interviews and fecal sampling. For Chachi participants, information was provided in their local language (chá palaa) by the community leader. For children’s participation, formal verbal consent was obtained from the parent/guardian. Personal data were anonymized and coded to maintain participant confidentiality. Interviews were conducted in a private and calm setting, typically at participants’ homes, ensuring respect for their privacy, property, belongings, and materials. For dog stool collection, no ethical authorization was required as no experimentation was conducted on the animals. Participants and dogs who tested positive for helminth eggs or larvae received appropriate single-dose anthelmintic treatment in accordance with the guidelines of the Ecuadorian Ministry of Public Health.

## 3. Results

### 3.1. Characteristic of the Study Population

A total of 300 inhabitants live in the two populations studied. Here, 86 adults (54 Chachi and 32 Montubios), including 59 females (68.6%) and 27 males (31.4%), aged from 17 to 86 years, responded the questionnaire. Among these participants, 67 were also interviewed in-depth. The respondents included women responsible for family care, community leaders, educators and health workers. A total of 273 participants, aged between 6 and 86 years, provided a fecal sample (187 from Chachi and 86 from Montubios communities). Additionally, 80 dog fecal samples were collected (44 from Chachi and 36 from Montubios communities).

### 3.2. Prevalence of Amphimerus spp. Infection

The average prevalence of infection in humans was 23%, with 21.9% in the Chachi and 25.6% in Montubios. Among those participants, 29% were female (42/145) and 16.4% were male (21/128) (Table 1).

The prevalence rate for the 80 dog samples averaged 16.2% with 15.9% in the Chachi communities and 16.7% in the Montubios communities.

### 3.3. Knowledge on Amphimeriasis Versus Sociodemographic and Educational Characteristics

A total of 68.6% (59/86) of respondents were not aware of amphimeriasis. Of the 86 questionnaires, 75 were completed by heads or representatives of families, and 11 by individuals from the health and educational sector. Females demonstrated greater familiarity with amphimeriasis (52%, 13/25) compared to males (23%, 14/61), with a *p*-value of 0.02. No significant differences were observed by age, living conditions, or access to electricity, although most participants live in their own houses (88.4%) and have electricity (98.8%). Differences were noted in the type of housing and cooking methods between the two population groups, with a predominance of wood stoves in the Chachi and gas stoves in the Montubios. These variations in lifestyle habits, influenced by cultural and socioeconomic factors, may be associated with the disease (Table 2). Of the 68.6% who were unaware of amphimeriasis, 96.6% (57/59) did not know if any community member had ever had the disease (*p*-value > 0.01). This variable did not show a significant difference between ethnic groups (*p*-value > 0.05). However, within the “infected/non-infected” subgroup, there was a significant association (*p*-value 0.03) with knowing a community member with amphimeriasis, although 83.9% (27/31) still responded as not knowing anyone with the disease.

Regarding education attainment, 93% (80/86) of participants possess reading and writing skills and have achieved basic, middle, or higher education levels. Our findings reveal no significant differences in knowledge about amphimeriasis between the Chachi and Montubios groups, or between infected and non-infected individuals (*p*-value > 0.05). Similarly, educational levels (elementary, secondary) did not show significant variations in terms of disease awareness (*p*-value = 0.3). However, 45.5% (5/11) of participants with tertiary education demonstrated greater awareness of the disease (Table 3).

### 3.4. Fish Consumption, Infection, and Knowledge of Amphimeriasis

Among the 86 respondents, 18 (20.8%) reported consuming raw fish and 51 (59.3%) reported consuming smoked fish. In the Chachi group, 16 (88.9%) consumed marinated fish with lime juice and salt, and 48 (94.1%) consumed smoked fish. In contrast, among the Montubios only 11.1% (2/18) reported consuming marinated fish, and 5.9% (3/51) consumed smoked fish. Regarding awareness of the disease and consumption of smoked fish, 63% of those knowledgeable about amphimeriasis consumed smoked fish, with an 85% prevalence among the Chachi (17/20) and 0% among the Montubios (0/7), demonstrating a significant association (*p*-value = 0.00). Additionally, 55.5% of those aware of the disease believed that consuming raw or pickled fish could be a route of transmission and infection, with 45% of the Chachi (9/20) and 85.6% of the Montubios (6/7) showing a significant association (*p*-value = 0.01) (Table 4).

Regarding reasons and preferences for fish preparations (raw, smoked, and dried), the analysis reveals significant differences between the Chachi and Montubios groups. The Chachi prefer smoked fish, with 39 participants favoring it due to family tradition and its superior flavor. In contrast, all Montubios reported not consuming smoked fish, and preferred raw fish in “curtido” preparations, citing better taste and cultural heritage as the primary reasons. Concerning dried fish, 32 participants preferred it for its better taste, while 18 cited other reasons, with the Chachi being the most frequent consumers (Table 4, Table 5 and Table 6).

### 3.5. Attitudes Related to Amphimeriasis

A total of 77.9% (67/86) of respondents were unaware of how easily liver fluke could be transmitted, showing a significant association between this lack of awareness and their perception of the disease (*p*-value = 0.04). All 86 participants expressed willingness to properly cook or fry fish and shrimps, with strong correlations between this behavior and their knowledge of amphimeriasis (*p*-value < 0.00), as well as with the ethnic groups (*p*-value = 0.01). Additionally, 39.5% (34/86) understood how to prevent liver fluke transmission, although this awareness was not significantly associated with disease knowledge (*p*-value = 0.15). However, a significant association was observed with ethnic groups (*p*-value < 0.00). All respondents (100%) stated they would properly cook fish and other potential intermediate hosts of the disease, including river shrimp (*Macrobrachium borellii*), bamboo shrimp (also known as “churo” or “gazapo” (*Aya scabra*)), and crabs or “pangoras” (*Hypolobocera aequatorialis*) (Table 7).

In the analysis of the “infected/non-infected” subgroup, a significant association was found between knowledge and the practice of defecating in the open field (*p*-value = 0.04). Moreover, within this subgroup, 56% (14/25) of the respondents were unaware that domestic animals can act as reservoirs of *Amphimerus* (*p*-value = 0.05). Statistically significant differences were observed between population groups in response to the question “Do you believe that the ‘liver fluke’ is easily transmitted among community members?” with a *p*-value < 0.05, regardless of the specific population group. Only 33.3% of respondents believed that the transmission of the parasite is easy. Regarding the question “How do you think the transmission of liver fluke can be prevented?”, six out of nine participants believed that transmission could be effectively prevented, with 50% of Chachi and 100% of Montubios holding this belief (Table 8).

### 3.6. Practices (Pets Care, Frequency and Preparation of Fish) Versus Knowledge of Amphimeriasis by Ethnic Groups

Questions regarding where food is consumed and participation in the preparation and consumption of fish at social events were highly significant (*p*-value < 0.01). In contrast, the relationship with ethnic groups showed significant differences in aspects such as species of fish consumed, frequency of consumption, methods of raw fish preparation, and pet management (*p*-value < 0.01). Among those aware of the disease, 40.7% (11/27) reported eating at home, compared to 71.2% (42/59) of those unaware of the disease. In the Chachi ethnic group, 65% reported consuming river fish weekly, compared to only 14.3% of the Montubios. Additionally, 42.9% of the Chachi and 90% of the Montubios reported preparing raw fish. Finally, 59.3% of those aware of amphimeriasis participated in the preparation and consumption of fish at social events, compared to 32.2% of those unaware.

Here, we found that 60% (12/20) of the Chachi feed their pets with fish or river animal remains, compared to 28.6% of the Montubios (*p*-value = 0.04). In terms of pet feces management, 50% of the Chachi do not collect pet feces, whereas 57% of the Montubios do not collect theirs (*p*-value = 0.01). Regarding pet deworming practices, 15% of the Chachi and 14.3% of Montubios deworm their dogs. No significant statistical associations were observed in the infected/non infected subgroup, nor were differences noted in relation to disease awareness or population group (Table 9).

Consumption preferences for fish and shrimp among the Chachi and Montubios demonstrate distinct patterns based on their awareness of amphimeriasis. Among the Chachi, 37.3% of those aware of the disease prefer shrimp, with other fish species preferred by percentages ranging from 1.9% to 22.2%; in contrast, a higher preference for shrimp at 62.7% is observed among those unaware. Both aware and unaware Montubios predominantly favor shrimp, with respective percentages of 18.5% and 81.5%. Overall, 90.7% (78/86) of participants consume shrimp, divided into 88.9% of those aware and 91.5% of those unaware of amphimeriasis. Regarding the preparation of raw fish or shrimp, 77.7% of Chachi and 28.1% of Montubios engage in this practice (Table 10).

In the qualitative analysis of 67 interviews (47 with Chachi, 20 with Montubios), two significant relationships were identified concerning knowledge, beliefs, and behaviors related to amphimeriasis. The first relationship pertains to the knowledge of amphimeriasis and its perceived risk. Independently of their familiarity with the disease, most participants recognized amphimeriasis as a severe condition, associating it with other hepatic diseases or intestinal parasitosis, and expressing concern over the distressing possibility of harboring a worm that could lead to disability or death. The second relationship concerns knowledge about potential modes of transmission, shaped by prior understanding of waterborne and foodborne infectious diseases, which influences perceptions about contamination, disinfection, and food preparation practices. Despite awareness of the risks associated with consuming undercooked fish and shrimps, the prevalent practice of consuming these foods pickled with lime juice or smoked—methods of preparation rather than cooking—highlights a disconnect between risk understanding and actual food practices, underscoring a critical need for public health education and intervention. During the in-depth interviews, we identified that several participants declined to participate for religious reasons, stating “We are with God, and we do not need medication for any of our illnesses”.

## 4. Discussion

This is the first study on amphimeriasis (*Amphimerus* liver fluke infection) to evaluate knowledge, attitudes and practices (KAP) and sociocultural determinants, conducted in the two ethnic populations recognized as endemic, located in the rural communities of the tropical coastal region of the Pacific Ocean of Ecuador [13]. Given the high prevalence of infection encountered in the present study, amphimeriasis should be considered as a public health problem in the populations studied. Human infections with *Amphimerus* were first reported in Ecuador in 2011 [11,12,13,14,21]. Our findings help bridge the knowledge gap regarding perceptions and KAP related to *Amphimerus* infection. Through a mixed-methods research design, we identified the following.

(1) The knowledge on the epidemiology of amphimeriasis among both ethnic groups was limited, with only 31.4% of participants having heard of the “liver worm”. There was no statistically significant difference in awareness between the Chachi (37%) and Montubio (21.9%) groups. The analysis further revealed that gender and age were significant factors affecting awareness; females (52%) were more knowledgeable than males (23%). Additionally, ages impacted knowledge differently across gender: younger females were more informed than younger males, whereas older males had greater awareness than older females. Younger participants, more likely to have secondary education, would have easier access to information via mobile phones or the internet. However, amphimeriasis is not yet covered in public media. We believe that those who are aware gained their knowledge from our previous research visits, which included investigations of *Amphimerus* and treatments of previously identified cases [11,12,13,21]. We also found that individuals both infected by and aware of the “liver worm” often consumed raw fish in popular preparations like “curtido” and smoked but are uninformed that these methods involve raw fish. Most participants identified raw fish as a potential transmission route for the “liver worm”. Unexpectedly, there was no difference in knowledge between the infected and non-infected groups, indicating a general lack of awareness. The practice of defecating in open fields was significantly more prevalent among the infected group (*p*-value = 0.04). Additionally, a considerable lack of knowledge was also found among health personnel and educators, including medical doctors and nurses working in nearby health centers and schools. Regarding symptom recognitions, 95% of the participants failed to identify any symptoms. This level of unawareness is high compared to studies on clonorchiasis and opisthorchiasis in Asia, where about 50% were unaware of the symptoms [8,22,23]. This finding underscores the importance of amphimeriasis as a public health issue and is particularly concerning given the lack of clinical signs, symptoms or pathological evidence in clinical consultations or diagnoses. Additionally, the absence of studies on amphimeriasis in the region contributes to a significant knowledge gap among health professionals and the general population. Therefore, there is a pressing need for clinical studies to address this gap in the future.

(2) The attitudes towards the prevention and treatment of *Amphimerus* infection were generally positive; all participants expressed a willingness to properly cook fish and other river crustaceans. This positive attitude is particularly significant given the importance of amphimeriasis as a public health problem. Fifty percent of Chachi and all Montubios believed that prevention could be easily achieved by thoroughly cooking river foods. Despite 68.8% of respondents being unaware of the disease, they were willing to undergo medication if diagnosed. This positive attitude may stem from the perception that liver diseases are severe. The attitudes of the Chachi were more positive than those of the Montubios in some aspects; indeed, 62.8% (54/86) of Chachi participants expressed willingness to participate in diagnostic and treatment programs. The variations in attitudes can be attributed to previous studies conducted by our group and others on different diseases in the Chachi. The Chachi benefit from community health workers trained by non-governmental organizations (NGOs), who aid in their understanding of liver fluke infection. The findings indicate that greater knowledge correlates with more positive attitudes towards disease management.

(3) The proportion of locals engaging in good practices to prevent the transmission of *Amphimerus* was low. This is especially concerning given that humans and dogs showed a high prevalence of infection, and good practices underscore the challenges in reducing transmission in endemic areas. Notably, 90% of Montubios and 42.9% of Chachi participants consume raw fish, with Chachi preferring smoked preparations and Montubios favouring “curtido”. Additionally, most respondents were unaware that domestic animals can serve as reservoirs of the “liver worm”. The poor practices identified were (1) feeding pets with fish remains, and (2) allowing dogs and cats to roam freely in the communities. Dogs and cats are recognized as reservoirs and are highly susceptible to *Amphimerus* infection [12]. We also identified that toilets and manholes may drain into rivers and streams, particularly during floods, or worse, discharge wastewater directly into these water bodies. This poor sanitation facilitates the contamination of water with *Amphimerus* eggs from infected individuals, perpetuating the parasite’s lifecycle in rivers water. A significant knowledge–practice gap exists between the awareness of transmission via contaminated food and the actual consumption practices involving raw fish and shrimp. Given the difficulty of changing food practices, addressing this cultural habit requires complex solutions.

(4) The habits of consuming raw fish were not solely due to a lack of knowledge but also influenced by taste, cultural practices, preservation methods, and misconceptions regarding the antimicrobial and disinfectant properties of lime juice. These entrenched dietary habits contributed to the persistent risk of transmission. Descriptive analyses of practices related to amphimeriasis show that certain dietary habits are consistent between individuals aware and unaware of the disease. However, aspects such as where food is consumed and participation in the preparation and consumption of fish at social events were found to be highly significant, indicating a strong association with disease awareness. In contrast, significant differences were observed in relationship with different groups, affecting variables such as the types of fish consumed, the frequency of consumption, methods of raw fish preparation, and pet management.

(5) Interestingly, the level of education did not significantly influence knowledge about amphimeriasis in this study. Research on liver flukes [24], schistosomiasis and clonorchiasis by Assefa et al. [25] and Vinh et al. [26] demonstrated that education plays a crucial role in enhancing disease awareness. Similarly, for other foodborne trematodiases, education has been identified as a key factor in increasing knowledge and promoting preventive practices against transmission and infection [22,27,28].

Taken together, the identified gaps in KAP among the participants will likely contribute to the continuous transmission and endemicity of *Amphimerus* infection in both ethnic groups. As this is the inaugural study of its kind, direct comparisons with previous studies on *Amphimerus* are not possible. The low levels of KAP observed in this study are similar to those reported for other fish-borne *Opisthorchiidae* infections, such as *Opisthorchis* in Thailand and Vietnam [8,24].

(6) We observed differences in the KAP results between the Chachi and Montubios groups. The frequency of fish consumption varied significantly, with 65% of Chachi eating fish weekly, compared to only 14.3% of Montubios. This difference is attributed to the availability of fish, as Chachi communities are situated alongside large rivers, whereas Montubios reside along streams. Additionally, fishing habits differ between the groups; Chachi typically fish nearly every day throughout the year, while Montubios fish primarily during the dry season. Notable differences were also observed in fish preparation methods, with 94.1% (48/54) of Chachi consuming smoked fish, in contrast to only 9.4% (3/32) of Montubios. Another important risk factor difference was animal feeding practices. Here, we found that 60% of Chachi feed their pets with fish or river animal remains, compared to 28.6% of the Montubios. Local fish were demonstrated to be highly infected with the metacercaria infective form [13]. This bad practice will maintain the lifecycle of *Amphimerus* and consequent endemicity. Proximity to streams and rivers plays a significant role, as these bodies of water are consistent sources of fish and crustaceans. Without enhancing knowledge and practices via health education and health promotion, the cycle of *Amphimerus* and infections is likely to persist indefinitely.

Although only 20.8% of locals stated consuming raw fish, many consumed raw or undercooked fish through popular dishes such as “curtido” and smoked preparations. They believed that “curtido” or marinated and smoked fish, crustaceans, and all kinds of meat are safe. During the interviews, it was evident that in both Chachi and Montubios, females are primarily responsible for food preparation. Therefore, we recommended directing educational interventions towards females to enhance safe food practices.

Interestingly, the two studied sites are in the rural tropical coastal region of Ecuador; however, Chachi communities are in the humid, hot, northern rainforest, while Montubios are in the hot, dry, mountainous center (see map in Figure 1). Consequently, rivers are perennial in the Chachi region, while Montubios communities are characterized by seasonal streams. Nevertheless, the prevalence of human infection was similar (Chachi 21.9%, Montubios 25.6%), along with comparable infections rate in dogs (Chachi 15.9%, Montubios 16.7%). This suggests that the intensity of infection in both domestic and sylvatic reservoirs may be comparable, warranting further investigation. Another contributing factor might be the long lifespan of liver flukes, which allows infections to persist once a host is infected. It is important to note that the prevalence rates in humans and dogs remain high, consistent with previous studies [13,21,29]. To address this, dogs and cats should be dewormed during rabies vaccination campaigns, and veterinarians should be informed and involved in controlling amphimeriasis.

Importantly, 95.3% of participants consume raw shrimps in “ceviche” preparations. Given the high prevalence of *Amphimerus* infection, it is crucial to investigate whether these crustaceans could act as secondary intermediate hosts for *Amphimerus*. Currently, it remains unknown if shrimp can be a reservoir of this trematode, although the existing literature has established that shrimps can transmit the lung fluke *Paragonimus* spp. [30]. In the study areas, we captured other crustaceans, including river shrimp (*Macrobrachium borellii*), bamboo shrimp (*Aya scabra*), and crabs (*Hypolobocera aequatorialis*). This line of inquiry is promising for determining whether these river crustaceans could also facilitate the transmission of *Amphimerus*.

In the qualitative analysis, the perceived seriousness of the “liver worm” was high, with participants often associating it with other hepatic diseases or intestinal parasitosis and expressing concern about harboring a worm internally that could lead to disability or death. Participants raised concerns about modes of transmission, influenced by their prior knowledge of waterborne and foodborne infections, as well as practices related to contamination, disinfection, and food preparation. This discrepancy underscores a disconnect between risk understanding and actual food preparation habits, highlighting a critical area for public health education and intervention.

A notable and intriguing finding was that, even after the objectives of the study were explained, some individuals declined to participate, stating, “We do not need human intervention in our health because we are in the hands of God”. This also highlights the importance of amphimeriasis as a public health issue, as it underscores how unsafe practices in the studied rural communities are influenced not only by a lack of knowledge and attitudinal factors, but also by deeply ingrained religious beliefs. These beliefs can act as significant barriers to effective health interventions, thereby perpetuating the risk of infection, necessitating the addressing of amphimeriasis in a One Heath context.

The present study had several limitations. (1) The formalin–ether sedimentation method (Ritchie) was used to assess the prevalence of infection. Although this method is effective, a combination of methods, including the Kato–Katz technique, is known to be more sensitive [21]; therefore, the prevalence of *Amphimerus* infection among residents and dogs may have been underestimated and could potentially be higher. However, we utilized the Ritchie method due to its lower cost and because fecal samples could not be processed in the field. (2) The number of participants might seem small; however, we surveyed 86 (28.7%), and conducted 67 (22.3%) interviews, from an estimated total population of around 300, representing approximately one-fourth of the inhabitants. (3) The limited number of participants was primarily due to concerns about contracting COVID-19 from city visitors and the ongoing armed insecurity in Ecuador, in addition to religious reasons, as previously mentioned. (4) We did not investigate whether the habit of consuming raw fish was influenced by economic conditions and social structures, an area that warrants further investigation. However, the significant strength of this study lies in its integrated approach, combining quantitative and qualitative methods. Furthermore, we obtained and compared the results from the two known endemic sites, inhabited by distinct ethnic groups. Given the importance of amphimeriasis as a public health problem, this study is particularly valuable as the first study to explore KAP related to *Amphimerus* infections. It has provided insights into both the facilitators and barriers relevant to the implementation of prevention and control strategies. Previous research has demonstrated that post-educational interventions significantly enhance knowledge and attitudes towards foodborne trematodiasis management [22,27].

## 5. Conclusions

Amphimeriasis was identified as a zoonotic and neglected public health problem in Ecuador. Despite previous diagnostic and treatment efforts by our research group in both populations, the prevalence of *Amphimerus* infection among humans and dogs across the two ethnic groups and study sites is still high. Knowledge remains limited, and practices are poor regarding the epidemiology of *Amphimerus* infection. However, a positive aspect of the study was the generally favorable attitudes observed. The findings indicate that a lack of knowledge is not the sole factor contributing to a habit of consuming raw fish; misunderstandings and cultural beliefs also play significant roles in the endemicity of amphimeriasis. Therefore, an integrated control strategy is essential for preventing and managing amphimeriasis in the studied populations. This strategy should include health education focused on safe fish preparation practices, along with the diagnosis and treatment of reservoir hosts, including dogs. Consideration should be given to mass drug administration using praziquantel for both humans and animals. Unfortunately, praziquantel for human use is currently unavailable in Ecuador. It is crucial that these interventions be culturally sensitive and tailored to local practices and beliefs to ensure their acceptance and effectiveness.

## Figures and Tables

**Figure 1 tropicalmed-09-00248-f001:**
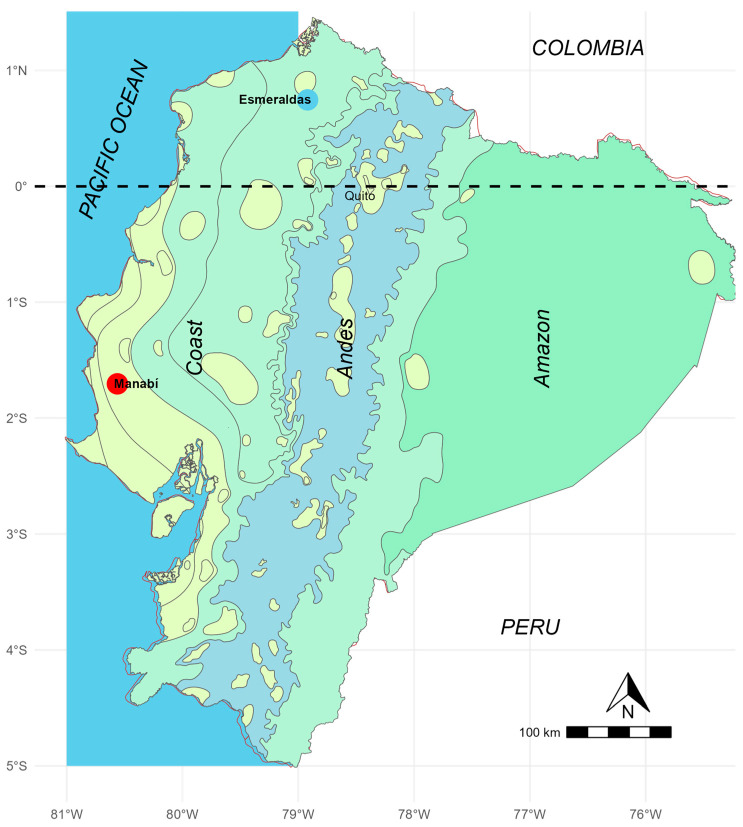
Climate map of Ecuador. The Chachi communities are located within a humid tropical climate zone, part of the Choco region (blue circle, Esmeraldas), while the Montubios site is situated in a dry tropical climate zone (red circle, Manabi).

**Figure 2 tropicalmed-09-00248-f002:**
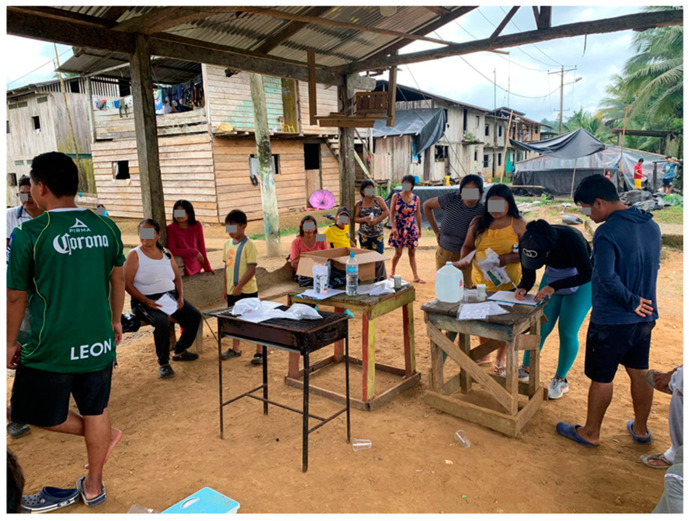
Chachi people during the application of questionnaires in a community located alongside of Cayapas river.

**Figure 3 tropicalmed-09-00248-f003:**
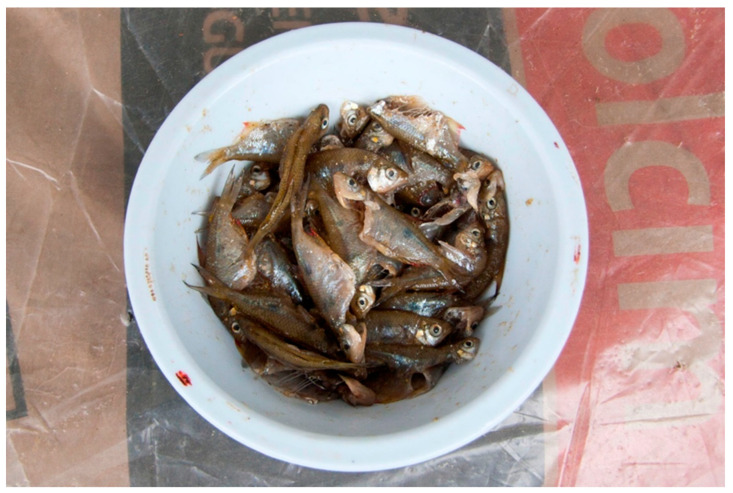
A popular Montubio’s dish, “curtido”, prepared with raw freshwater fish, lime juice, and salt. The dish is ready to eat with rice after marinating for two hours or more. The photographed dish was shared with us by community members.

**Table 1 tropicalmed-09-00248-t001:** *Amphimerus* infection prevalence in humans, overall and by ethnic group and gender.

Infection	Total n (%)	Ethnic Group		Gender	
		Chachi n (%)	Montubios n (%)	Male n (%)	Female n (%)
Positives	63 (23.1)	41 (21.9)	22 (25.6)	21 (16.4)	42 (29)
Negatives	210 (76.9)	146 (78.1)	64 (74.4)	107 (83.6)	103 (71)
Total	273 (100)	187 (100)	86 (100)	128 (100)	145 (100)

**Table 2 tropicalmed-09-00248-t002:** Sociodemographic characteristics and ethnicity vs. knowledge of amphimeriasis.

Sociodemographic	Total	Knowledge of Amphimeriasis			Ethnic Group		
	n (%)	Yes n (%)	No n (%)	*p*-Value	Chachi n (%)	Montubios n (%)	*p*-Value
What is your gender?							
Male	61 (70.9)	14 (51.9)	47 (79.7)	0.02 *	13 (65.0)	1 (14.3)	0.06
Female	25 (29.1)	13 (48.2)	12 (20.3)		7 (35.0)	6 (85.7)	
Total	86	27 (31.4)	59 (68.6)		20 (100.0)	7 (100.0)	
How old are you?							
Youngs <20	2 (2.3)	0 (0.0)	2 (3.4)	0.61	0 (0.0)	0 (0.0)	0.26
Adults (20–40)	39 (45.3)	12 (44.4)	27 (45.8)		8 (40.0)	4 (57.1)	
Oldest >41	45 (52.3)	15 (55.6)	30 (50.9)		12 (60.0)	3 (42.9)	
Total		27 (31.4)	59 (68.6)		20 (100.0)	7 (100.0)	
How is your house?							
Rented	3 (3.5)	1 (3.7)	2 (3.4)	0.98	0 (0.0)	1 (14.3)	0.01 *
Borrowed	7 (8.1)	2 (7.4)	5 (8.5)		0 (0.0)	2 (28.6)	
Owned	76 (88.4)	24 (88.9)	52 (88.1)		20 (100.0)	4 (57.1)	
Total		27 (31.4)	59 (68.6)		20 (100.0)	7 (100.0)	
Do you have kitchen? gas, firewood and mix (gas–firewood)							
Gas	34 (39.5)	13 (48.2)	21 (35.6)	0.29	6 (30.0)	7 (100)	0.01 *
Firewood	9 (10.5)	1 (3.7)	8 (13.6)		1 (5.0)	0 (0.0)	
Mixed	43 (50.0)	13 (48.2)	30 (50.9)		13 (65.0)	0 (0.0)	
Total		27 (31.4)	59 (68.6)		20 (100.0)	7 (100.0)	
Do you have electricity?							
No	1 (1.2)	(0.0)	1 (1.7)	0.34	0 (0.0)	(0.0)	0.01 *
Yes	85 (98.8)	27 (100.0)	58 (98.3)		20 (100.0)	7 (100.0)	
Total		27 (31.4)	59 (68.6)		20 (100.0)	7 (100.0)	

* significance.

**Table 3 tropicalmed-09-00248-t003:** Educational characteristics and ethnicity versus knowledge of amphimeriasis.

Education	Total	Knowledge of Amphimeriasis			Ethnic Group		
	n (%) ^1^	Yes n (%)	No n (%)	*p*-Value	Chachi n (%)	Montubios n (%)	*p*-Value
Do you know how to read and write?							
No	6 (7.0)	3 (11.1)	3 (5.1)	0.57	2 (10.0)	1 (14.3)	1
Yes	80 (93.0)	24 (88.9)	56 (94.9)		18 (90.0)	6 (85.7)	
Total		27 (31.4)	59 (68.6)		20 (100.0)	7 (100.0)	
Education level: Primary, secondary, higher, NA (no answer)							
NA	6 (7.0)	3 (11.1)	3 (5.1)	0.32	2 (10.0)	1 (14.3)	0.16
Primary	33 (38.4)	11 (40.7)	22 (37.3)		6 (30.0)	5 (71.4)	
Secondary	36 (41.9)	8 (29.6)	28 (47.5)		8 (40.0)	0 (0.0)	
Higher	11 (12.8)	5 (18.5)	6 (10.2)		4 (20.0)	1 (14.3)	
Total		24 (27.9)	56 (65.1)		20 (100.0)	7 (100.0)	

^1^ number and percentage of participants.

**Table 4 tropicalmed-09-00248-t004:** Knowledge of amphimeriasis and fish consumption among ethnic groups.

Knowledge	Total	Knowledge of Amphimeriasis			Ethnic Group		
	n (%)	Yes n (%)	No n (%)	*p*-Value	Chachi n (%)	Montubios n (%)	*p*-Value
Have you heard of or are you familiar with liver fluke disease?							
No	59 (68.6)	0	0	0.00 *	0	0	0.00 *
Yes	27 (31.4)	27 (100.0)	59 (100.0)		20 (100.0)	7 (100.0)	
Total		27 (31.4)	59 (68.6)		20 (100.0)	7 (100.0)	
Do you know anyone (in the community) who may have or have had liver fluke?							
No	49 (57.0)	12 (44.4)	37 (62.7)	0.00 *	9 (45.0)	3 (42.9)	0.70
Don’t know	25 (29.1)	5 (18.5)	20 (33.9)		3 (15.0)	2 (28.6)	
Yes	12 (14.0)	10 (37.0)	2 (3.4)		8 (40.0)	2 (28.6)	
Total		27 (31.4)	59 (68.6)		20 (100.0)	7 (100.0)	
Do you consume smoked fish? Reasons: Tastier, more nutritious, family tradition, preservation							
No	37 (43.0)	10 (37.0)	27 (45.8)	0.60	3 (15.0)	7 (100.0)	0.00 *
Yes	49 (57.0)	17 (63.0)	32 (54.2)		17 (85.0)	0 (0.0)	
Total		27 (31.4)	59 (68.6)		20 (100.0)	7 (100.0)	
Do you believe that raw or pickled fish could infect you with liver fluke?							
No	31 (36.1)	11 (40.7)	20 (33.9)	0.55	11 (55.0)	0 (0.0)	0.01 *
Don’t know	7 (8.1)	1 (3.7)	6 (10.2)		(0.0)	1 (14.3)	
Yes	48 (55.8)	15 (55.6)	33 (55.9)		9 (45.0)	6 (85.7)	
Total		27 (31.4)	59 (68.6)		20 (100.0)	7 (100)	

* significance.

**Table 5 tropicalmed-09-00248-t005:** Knowledge of amphimeriasis and fish consumption in infected/non infected groups.

Knowledge	Total	Awareness of Amphimeriasis		
	n (%)	Yes n (%)	No n (%)	*p*-Value
Do you know anyone (in the community) who may have or has had liver fluke?				
Yes	5 (16.1)	4 (44.4)	1 (4.6)	0.03 *
Don’t know	8 (25.8)	1 (11.1)	7 (31.8)	
No	18 (58.1)	4 (44.4)	14 (63.6)	
Total	31 (100)	9 (100)	22 (100)	
Do you eat smoked fish?				
Yes	11 (35.5)	6 (66.7)	5 (22.7)	0.04 *
No	20 (64.5)	3 (33.3)	17 (73,.3)	
Total	31 (100)	9 (100)	22 (100)	

* significance.

**Table 6 tropicalmed-09-00248-t006:** Reasons and preferences for fish preparations (raw, smoked, dried) by ethnic groups.

Preferences	Raw Fish			Smoked Fish			Dried Fish		
	Chachi	Montubio	Total	Chachi	Montubio	Total	Chachi	Montubio	Total
More delicious/tasty	1	1	2	36	3	39	22	10	32
More nutritious				2	0	2	1	1	2
Family tradition	8	1	9						
Preservation							8	2	10
All preferences	7	0	7	10		10	16	2	18
Do not consume	38	30	68	6	29	35	7	17	24
Total	54	32	86	54	32	86	54	32	86

**Table 7 tropicalmed-09-00248-t007:** Attitudes of all participants regarding the cooking of fish and crustaceans, overall and by ethnicity.

Attitudes	Total	Knowledge of Amphimeriasis			Ethnic Group		
	n (%)	Yes n (%)	No n (%)	*p*-Value	Chachi n (%)	Montubios n (%)	*p*-Value
Do you believe that liver fluke is easily transmitted among community members?							
Don’t know	67 (77.9)	17 (63.0)	50 (84.7)	0.04 *	13 (65%)	4 (57.1%)	0.36
Yes	19 (22.1)	10 (37.0)	9 (15.3)		7 (35%)	3 (42.9%)	
Total		27 (31.4)	59 (68.6)		20 (100%)	7 (100%)	
Would you be willing to cook or fry your fish, snails, or crabs thoroughly?							
Yes	86 (100.0)	27 (100.0)	59 (100.0)	0.00 *	20 (100.0)	7 (100.0)	0.01 *
How do you think liver fluke infection can be prevented?							
Don’t know	24 (27.9)	5 (18.5)	19 (32.2)	0.15	4 (20%)	1 (14.3%)	0.00 *
Have an idea	32 (37.2)	14 (51.9)	18 (30.5)		14 (70%)	0 (0%)	
Know	30 (34.9)	8 (29.6)	22 (37.3)		2 (10%)	6 (85.7%)	
Total		27 (31.4)	59 (68.6)		20 (100%)	7 (100%)	

* significance.

**Table 8 tropicalmed-09-00248-t008:** Beliefs of the infected/not infected groups regarding amphimeriasis.

Attitudes	Total	Knowledge of Amphimeriais		
	n (%)	Yes n (%)	No n (%)	*p*-Value
Do you believe that liver fluke is easily transmitted among community members?				
Yes	5 (16.1)	3 (33.3)	2 (9.1)	0.05 *
No	3 (9.7)	2 (22.2)	1 (4.6)	
Don’t know	23 (74.2)	4 (44.4)	119 (63.6)	
Total	31 (100)	9 (100)	22 (86.4)	
Do you believe that defecating in the wild contaminates the water?				
Yes	17 (54.8)	8 (88.9)	9 (40.9)	0.04 *
Don’t know	7 (22.6)	1 (11.1)	6 (27.3)	
No	7 (22.6)	0 (0)	7 (31.8)	
Total	31 (100)	9 (100)	22 (100)	
Do you believe that your dogs and cats can have liver fluke?				
Yes	11 (44)	6 (87.7)	5 (27.9)	0.05 *
Don’t Know	11 (44)	1 (14.3)	10 (55.6)	
No	3 (12)	0 (0)	3 (31.8)	
Total	25 (100)	7 (100)	18 (100)	

* significance.

**Table 9 tropicalmed-09-00248-t009:** Practices of participants vs. knowledge of amphimeriasis, overall and by ethnic groups.

Practices	Total	Awareness of Amphimeriasis			Ethnic Group		
	n (%)	Yes n (%)	No n (%)	*p*-Value	Chachi n (%)	Montubios n (%)	*p*-Value
Where do you eat your meals?							
Home	53 (61.6)	11 (40.7)	42 (71.2)	0.01 *	6 (30)	5 (71.4)	0.14
Home/Outside	33 (38.4)	16 (59.3)	17 (28.8)		14 (70)	2 (28.6)	
Total		27 (31.4)	59 (68.6)		20 (100)	7 (100)	
What types of fish do you eat? “viejas” (*Cichlasoma festae*, *Parachromis managuensis*), “guanchiche” (*Hoplias malabaricus*, *Hoplias microlepis*), “sardinas” (*Triportheus angulatus*, *Hemigrammus unilineatus*), “engordas” (*Colossoma macropomum*, *Piaractus brachypomus*), “anchas” (*Brycon* spp., *Brycon cephalus*)							
At least one	85 (98.8)	27 (100)	58 (98.3)	1	20 (100)	7 (100)	0.012 *
Others	1 (1.2)	0 (14.8)	1 (1.7)		0 (0)	0 (0)	
Total		27 (31.4)	59 (68.6)		20 (100)	7 (100)	
How often do you eat river fish?							
Daily	9 (10.5)	2 (7.4)	7 (11.9)	0.56	2 (10)	0 (0)	0.04 *
Monthly	9 (10.5)	1 (3.7)	8 (13.6)		1 (5)	0 (0)	
Never	4 (4.7)	2 (7.4)	2 (3.4)		1 (5)	1 (14.3)	
Weekly	41 (47.7)	14 (51.9)	27 (45.8)		13 (65)	1 (14.3)	
Seasonally	23 (26.7)	8 (29.6)	15 (25.4)		3 (15)	5 (71.4)	
Total		27 (31.4)	59 (68.6)		20 (100)	7 (100)	
How do you prepare or consume the fish?							
cooked	28 (32.6)	6 (22.2)	22 (37.3)	0.25	2 (10)	4 (57.1)	0.04 *
Include raw	58 (67.4)	21 (77.8)	37 (62.7)		18 (90)	3 (42.9)	
Total		27 (31.4)	59 (68.6)		20 (100)	7 (100)	
Do you or your neighbors or family prepare and consume fish at social events/parties?							
No	51 (59.3)	11 (40.7)	40 (67.8)	0.03 *	7 (35)	4 (57.1)	0.56
Yes	35 (40.7)	16 (59.3)	19 (32.2)		13 (65)	3 (42.9)	
Total		27 (31.4)	59 (68.6)		20 (100)	7 (100)	
Do you or any member of your family feed river fish or animal remains to dogs and cats?							
Don’t know	9 (10.5)	2 (7.4)	7 (11.9)	0.49	4 (20)	0 (0)	0.04 *
No	9 (10.5)	1 (3.7)	8 (13.6)		4 (20)	5 (71.4)	
Yes	4 (4.7)	2 (7.4)	2 (3.4)		12 (60)	2 (28.6)	
Total		27 (31.4)	59 (68.6)		20 (100)	7 (100)	
What do you do with the feces of dogs and cats?							
Nothing	9 (10.5)	2 (7.4)	7 (11.9)	0.48	10 (50)	4 (57.1)	0.002 *
No pets	9 (10.5)	1 (3.7)	8 (13.6)		0 (0)	3 (42.9)	
Collect	4 (4.7)	2 (7.4)	2 (3.4)		10 (50)	0 (0)	
Total		27 (31.4)	59 (68.6)		20 (100)	7 (100)	
Have your pets received treatment for parasites?							
Don’t know	9 (10.5)	2 (7.4)	7 (11.9)	0.15	7 (35)	6 (85.7)	0.04 *
No	9 (10.5)	1 (3.7)	8 (13.6)		10 (50)	0 (0)	
Yes	4 (4.7)	2 (7.4)	2 (3.4)		3 (15)	1 (14.3)	
Total		27 (31.4)	59 (68.6)		20 (100)	7 (100)	

* significance.

**Table 10 tropicalmed-09-00248-t010:** Consumption of raw or cooked fish and shrimp among aware and unaware individuals, categorized by ethnic group.

Type of Preparation	Chachi n (%)				Montubios n (%)			
	Raw		Cooked		Raw		Cooked	
	Yes	No	Yes	No	Yes	No	Yes	No
With shrimp/fish	16 (29.6)	26 (48.1)	2 (3.7)	4 (7.4)	2 (6.2)	7 (21.9)	3 (9.4)	15 (46.9)
Without shrimp/fish	2 (3.7)	3 (5.6)	0 (0)	1 (1.9)	1 (3.1)	1 (3.1)	1 (3.1)	2 (6.2)
Total	18	29	2	5	3	8	4	17

## Data Availability

Data will be made available as requested.

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
