# Peer review of "Amphimeriasis in Ecuador—Prevalence, Knowledge, and Socio-Cultural Practices Among Indigenous Chachi and Montubios Populations: A Mixed-Methods Cross-Sectional Study"

_tropicalmed, 2024, doi:10.3390/tropicalmed9100248_

Round 1
Reviewer 1 Report
Comments and Suggestions for Authors
The main question addressed by the research is KAP, I do not see the crucial outcome. The topic is relevant to a lot of KAP with Opisthorchiasis. Something specific behavior for prevention or control to strategy compared with other published material. The conclusions are correct according to objective, but I do not see new knowledge or innovation.
Review with crucial gap and integration, what is the good prevention and control ?
Regression analysis for predictive model of prevalence
Comments on the Quality of English Languageno commemt
Author Response
Dear Reviewer,
Thanks for the comment. First, it is important to highlight that this study is the first to investigate KAP related to amphimeriasis in a country where human infections were first reported globally. A key outcome of the study is the identification of a significant gap in KAP regarding amphimeriasis among the Chachi and Montubios populations, despite the high prevalence of Amphimerus infections (23% in humans and 16.2% in dogs). Furthermore, the study provides crucial insights by showing that traditional dietary practices, particularly the consumption of raw (“curtido and ceviche”) and smoked fish, are major drivers of transmission, with these behaviors deeply rooted in cultural and social contexts, including religion. Additionally, it highlights a lack of awareness about the disease among local health personnel and academics, emphasizing the urgent need for targeted, culturally sensitive health education interventions. This research underscores the critical role of engaging community leaders, including religious figures, in addressing this neglected public health threat. In response to the reviewer comments, we have performed and included the statistical analysis for all variables and discussed these findings throughout the manuscript.
Reviewer 2 Report
Comments and Suggestions for Authors
This manuscript addresses a relatively unknown parasitic disease, Amphimeriasis, obtained through eating raw or undercooked fish. Human infections were first reported in Ecuador in 2011. The genus Amphimerus is known to infect mammals from the Americas, including Canada, the United States, Costa Rica, Panama, Colombia, Brazil, and Peru, and therefore potentially human infections could be more wide-spread.
The manuscript reads very well and it was a pleasure reading it. I have only minor points for the authors to address (see annotated pdf file).

Author Response
Thank you to Reviewer 2 for the helpful feedback.
As requested by the reviewer, we have made minor revisions to scientific names and some formatting edits in the tables that are reflected in the revised version with tracked changes. We also corrected instances where a sentence began with a number. Additionally, we have removed the redundant phrase: “The KAP questionaires were coded according to the questions and segmented into categories: Geographical and Demographic (G_), Knowledge (C_), Attitudes (A_), and Practices (P_) for analytical purposes.”
In Table 9, we added the scientific name of each local fish name to make this information clear. ¿What types of fish do you eat? “viejas” (Cichlasoma festae, Parachromis managuensis). “guanchiche” (Hoplias malabaricus, Hoplias microlepis). “sardinas” (Triportheus angulatus, Hemigrammus unilineatus). “engordas” (Colossoma macropomum, Piaractus brachypomus). “anchas” (Brycon spp., Brycon cephalus)
We have rephrased the sentence, “Do you believe that the ‘liver fluke’ is easily transmitted among community members?” with a p-value < 0.05, irrespective of the population group”. The revised sentence now reads <<< Do you believe that the ‘liver fluke’ is easily transmitted among community members?” with a p-value < 0.05, regardless of the specific population group>>> (Line 402).
Reviewer 3 Report
Comments and Suggestions for Authors
in the Amphimerus Authors.docx

No problem with English language
Author Response
We appreciate your positive evaluation and your recommendation for publication in “Tropical Medicine and Infectious Disease” journal.
Thank you for your thoughtful and constructive feedback. We appreciate the Reviewer observations regarding our study, especially the point raised about the “One Health” approach (interaction between humans, animals, and environment) as we wrote in the conclusion section. We agree that this perspective is crucial for understanding the transmission dynamics of Amphimerus infection and endemicity. In our revised manuscript, we incorporate a more detailed discussion on the interactions among these factors to align with the “One Health” framework and emphasize its relevance to our integrated control strategy.
Regarding the high rate of parasitism in dogs, we acknowledge that this is a significant finding, confirming that amphimeriasis is a zoonosis. We recommend treating dogs during rabies vaccination campaigns, highlighting how leveraging existing veterinary interventions could be effective for controlling Amphimerus infections.
We also find your comments on the socio-economic conditions insightful. We discussed the possible influence of economic factors on dietary choices and how limited access to other food sources may impact the consumption of fish and other aquatic animals in these communities.
Your suggestion to explore whether shrimps could serve as intermediate hosts for Amphimerus is welcome!. We plan to highlight this as an area for further research and perform a study in the same researched communities. Thanks for that great idea!.
Reviewer 4 Report
Comments and Suggestions for Authors
The authors aimed at understanding the behaviour of 2 population groups in relation to Amphimerus epidemiology, seeking to implement effective intervention strategies for the prevention, control, and/or elimination of Amphimerus infection. They also microscopically analyzed stool samples in an attempt to update the infection prevalence. The research is important and highlights the public health importance of Amphimerus.
The authors did a wonderful job in bringing out the associations between the various variables that contribute to the prevalence of Amphimeriasis. This study goes on to show the various factors that could be utilised in the prevention measures of the disease. However, the authors did not adequately elaborate on the importance of amphimeriasis as a public health threat.
Comments on the Quality of English LanguageThe manuscript needs minor language edits
Author Response
Dear Reviewer,
Thank you for pointing out the importance of amphimeriasis as a public health issue. In the revised manuscript, we include a more detailed discussion on the significance of amphimeriasis as a public health concern, as we mentioned for Reviewer 1. We have made specific updates and rephrasing in the manuscript for your consideration as follow:
Amphimeriasis is a significant public health concern in Ecuador, as highlighted by the high prevalence of Amphimerus infection (23% in humans and 16.2% in dogs, demonstrated in this study) among indigenous Chachi and Montubios populations in tropical regions. The disease, primarily transmitted through the consumption of raw or undercooked river fish, poses a serious health risk, particularly in communities where traditional culinary practices are deeply ingrained. The study reveals that while the majority of the population recognizes the disease's severity, there is a widespread lack of knowledge about its transmission routes and prevention, even among health professionals. This underscores the potential for amphimeriasis to continue spreading unless targeted public health interventions, such as education campaigns that respect cultural norms, are implemented. Amphimeriasis represents not only a parasitic disease but also a broader challenge that requires addressing socio-cultural behaviors and strengthening local health systems to mitigate its impact on vulnerable populations.
Hopefully we have addressed correctly all comments of the Editor and Reviewers.
Warm regards
Manuel Calvopina (corresponding author)
Round 2
Reviewer 1 Report
Comments and Suggestions for Authors
Need more innovation and new knowledges for prevention and control